# Differentiable Sampling of Categorical Distributions Using the CatLog-Derivative Trick

**Lennert De Smet**
KU Leuven

**Emanuele Sansone**
KU Leuven

**Pedro Zuidberg Dos Martires**
Örebro University

## Abstract

Categorical random variables can faithfully represent the discrete and uncertain aspects of data as part of a discrete latent variable model. Learning in such models necessitates taking gradients with respect to the parameters of the categorical probability distributions, which is often intractable due to their combinatorial nature. A popular technique to estimate these otherwise intractable gradients is the Log-Derivative trick. This trick forms the basis of the well-known REINFORCE gradient estimator and its many extensions. While the Log-Derivative trick allows us to differentiate through samples drawn from categorical distributions, it does not take into account the discrete nature of the distribution itself. Our first contribution addresses this shortcoming by introducing the CatLog-Derivative trick– a variation of the Log-Derivative trick tailored towards categorical distributions. Secondly, we use the CatLog-Derivative trick to introduce IndeCateR, a novel and unbiased gradient estimator for the important case of products of independent categorical distributions with provably lower variance than REINFORCE. Thirdly, we empirically show that IndeCateR can be efficiently implemented and that its gradient estimates have significantly lower bias and variance for the same number of samples compared to the state of the art.

## 1 Introduction

Categorical random variables naturally emerge in many domains in AI, such as language modelling, reinforcement learning and neural-symbolic AI [10]. They are compelling because they can faithfully represent the discrete concepts present in data in a sound probabilistic fashion. Unfortunately, inference in probabilistic models with categorical latent variables is usually computationally intractable due to its combinatorial nature. This intractability often leads to the use of sampling-based, approximate inference techniques, which in turn poses problems to gradient-based learning as sampling is an inherently non-differentiable process.

In order to bypass this non-differentiability, two main classes of gradient estimators have been developed. On the one hand, there is a range of unbiased estimators based on the Log-Derivative trick and the subsequent REINFORCE gradient estimator [40]. On the other hand, we have biased estimators that use continuous relaxations to which the reparametrisation trick [33] can be applied, such as the Gumbel-Softmax trick [16, 22].

A clear advantage of the REINFORCE estimator over relaxation-based estimators is its unbiased nature. However, REINFORCE tends to be sample-inefficient and its gradient estimates exhibit high variance in practice. To resolve these issues, methods have been proposed that modify REINFORCE by, for instance, adding control variates [31, 37]. These modified estimators have been shown to deliver more robust gradient estimates than standard REINFORCE.

Instead of modifying REINFORCE, we take a different approach and modify the Log-Derivative trick by explicitly taking into account that we are working with multivariate categorical distributions.

37th Conference on Neural Information Processing Systems (NeurIPS 2023).

We call this first contribution the *CatLog-Derivative trick*. Interestingly, we show that our CatLog-Derivative trick leads to Rao-Blackwellised estimators [4], immediately giving us a guaranteed reduction in variance. The CatLog-Derivative trick can also be seen as a generalisation of the Local Expectation Gradients (LEG) [36] capable of exploiting more structural, distributional properties. This connection to LEG will be clarified throughout the paper.

As a second contribution, we propose IndeCateR (read as 'indicator'), a gradient estimator for the special case of independent categorical random variables. IndeCateR is a hyperparameter-free estimator that can be implemented efficiently by leveraging parallelisation on modern graphical processing units (GPUs). Thirdly, we empirically show that IndeCateR is competitive with comparable state-of-the-art gradient estimators on a range of standard benchmarks from the literature.

## 2 Notation and Preliminaries

Throughout this paper, we consider expectations with respect to multivariate categorical probability distributions of the form

$$\mathbb{E}_{\mathbf{X} \sim p(\mathbf{X})} [f(\mathbf{X})] = \sum_{\mathbf{x} \in \Omega(\mathbf{X})} p(\mathbf{x}) f(\mathbf{x}), \tag{2.1}$$

where we assume this expectation to be finite. The symbol $\mathbf{X}$ denotes a random vector $(X_1, \ldots, X_D)$ of $D$ categorical random variables while $p(\mathbf{X})$ denotes a multivariate probability distribution. The expression $\mathbf{X} \sim p(\mathbf{X})$ indicates that the random vector $\mathbf{X}$ is distributed according to $p(\mathbf{X})$. On the right-hand side of Equation (2.1) we write the expectation as an explicit sum over $\Omega(\mathbf{X})$, the finite sample space of the random vector $\mathbf{X}$, using $\mathbf{x} = (x_1, \ldots, x_D)$ for the specific assignments of the random vector $(X_1, \ldots, X_D)$.

Given an order of the random variables in $\mathbf{X}$, we can induce a factorisation of the joint probability distribution as follows

$$p(\mathbf{X}) = \prod_{d=1}^{D} p(X_d \mid \mathbf{X}_{<d}). \tag{2.2}$$

Here, $\mathbf{X}_{<d}$ denotes the ordered set of random variables $(X_1, \ldots, X_{d-1})$. Similarly, $\mathbf{X}_{>d}$ will denote the ordered set $(X_{d+1}, \ldots, X_D)$ in subsequent sections.

When performing gradient-based learning, we are interested in partial derivatives of the expected value in (2.1), i.e., $\partial_\lambda \mathbb{E}_{\mathbf{X} \sim p(\mathbf{X})} [f(\mathbf{X})]$. Here, we take the partial derivative of the expectation with respect to the parameter $\lambda$ and assume that the distribution $p(\mathbf{X})$ and the function $f(\mathbf{X})$ depend on a set of parameters $\Lambda$ with $\lambda \in \Lambda$. For probability distributions to which the reparametrisation trick does not apply, we can rewrite the partial derivative using the Log-Derivative trick.

**Theorem 2.1** (Log-Derivative Trick [40]). Let $p(\mathbf{X})$ be a probability distribution and $f(\mathbf{X})$ such that its expectation is finite, with both functions depending on a set of parameters $\Lambda$. Then, it holds that

$$\partial_\lambda \mathbb{E}_{\mathbf{X} \sim p(\mathbf{X})} [f(\mathbf{X})] = \mathbb{E}_{\mathbf{X} \sim p(\mathbf{X})} [\partial_\lambda f(\mathbf{X})] + \mathbb{E}_{\mathbf{X} \sim p(\mathbf{X})} [f(\mathbf{X}) \partial_\lambda \log p(\mathbf{X})]. \tag{2.3}$$

In general, both expectations in Equation (2.3) are intractable and often estimated with a Monte Carlo scheme. The most immediate such estimation is provided by the REINFORCE gradient estimator [40]

$$\partial_\lambda \mathbb{E}_{\mathbf{X} \sim p(\mathbf{X})} [f(\mathbf{X})] \approx \frac{1}{N} \sum_{n=1}^{N} \left( \partial_\lambda f(\mathbf{x}^{(n)}) + f(\mathbf{x}^{(n)}) \partial_\lambda \log p(\mathbf{x}^{(n)}) \right). \tag{2.4}$$

The superscript on $\mathbf{x}^{(n)}$ denotes that it is the $n^{\text{th}}$ sample vector drawn from $p(\mathbf{X})$.

A well-known problem with the REINFORCE gradient estimator is the high variance stemming from the second term in Equation (2.4). A growing body of research has been tackling this problem by proposing variance reduction techniques [11, 31, 37, 39]. In what follows we will focus on estimating this second term and drop the first term, since it can be assumed to be unproblematic.

# 3 The CatLog-Derivative Trick

The standard Log-Derivative trick and its corresponding gradient estimators are applicable to both discrete and continuous probability distributions. However, this generality limits their usefulness when it comes to purely categorical random variables. For example, the REINFORCE gradient estimator suffers from high variance when applied to problems involving high-dimensional multivariate categorical random variables. In such a setting there are exponentially many possible states to be sampled, which makes it increasingly unlikely that a specific state gets sampled. We now introduce the CatLog-Derivative trick that reduces the exponential number of states arising in a multivariate categorical distribution by exploiting the distribution's factorisation.

**Theorem 3.1** (CatLog-Derivative Trick). Let $p(\mathbf{X})$ be a multivariate categorical probability distribution that depends on a set of parameters $\Lambda$ and assume $\mathbb{E}_{\mathbf{X} \sim p(\mathbf{X})}[f(\mathbf{X})]$ is finite. Then, it holds that $\partial_\lambda \mathbb{E}_{\mathbf{X} \sim p(\mathbf{X})}[f(\mathbf{X})]$ is equal to

$$\sum_{d=1}^{D} \sum_{x_\delta \in \Omega(X_d)} \mathbb{E}_{\mathbf{X}_{<d} \sim p(\mathbf{X}_{<d})} \left[ \partial_\lambda p(x_\delta \mid \mathbf{X}_{<d}) \mathbb{E}_{\mathbf{X}_{>d} \sim p(\mathbf{X}_{>d}|x_\delta, \mathbf{X}_{<d})}[f(\mathbf{X}_{\neq d}, x_\delta)] \right] \quad (3.1)$$

*Proof.* We start by applying the standard Log-Derivative trick and fill in the product form of the categorical distribution followed by pulling this product out of the logarithm and the expectation

$$\mathbb{E}_{\mathbf{X} \sim p(\mathbf{X})}[f(\mathbf{X}) \partial_\lambda \log p(\mathbf{X})] = \mathbb{E}_{\mathbf{X} \sim p(\mathbf{X})} \left[ f(\mathbf{X}) \partial_\lambda \log \prod_{d=1}^{D} p(X_d \mid \mathbf{X}_{<d}) \right], \quad (3.2)$$

$$= \sum_{d=1}^{D} \mathbb{E}_{\mathbf{X} \sim p(\mathbf{X})}[f(\mathbf{X}) \partial_\lambda \log p(X_d \mid \mathbf{X}_{<d})]. \quad (3.3)$$

To continue, we write out the expectation explicitly and write the sum for the random variable $X_d$ separately, resulting in

$$\sum_{d=1}^{D} \sum_{\mathbf{x} \in \Omega(\mathbf{X}_{\neq d})} \sum_{x_\delta \in \Omega(X_d)} p(\mathbf{x}_{\neq d}, x_\delta) f(\mathbf{x}_{\neq d}, x_\delta) \partial_\lambda \log p(x_\delta \mid \mathbf{x}_{<d}). \quad (3.4)$$

Next, we factorize the joint probability $p(\mathbf{x}_{\neq d}, x_\delta)$ as $p(\mathbf{x}_{>d} \mid x_\delta, \mathbf{x}_{<d}) p(x_\delta \mid \mathbf{x}_{<d}) p(\mathbf{x}_{<d})$. Multiplying the second of these factors with $\partial_\lambda \log p(x_\delta \mid \mathbf{x}_{<d})$ gives us $\partial_\lambda p(x_\delta \mid \mathbf{x}_{<d})$. Finally, plugging $\partial_\lambda p(x_\delta \mid \mathbf{x}_{<d})$ into Equation (3.4) gives the desired expression. □

Intuitively, the CatLog-Derivative trick decomposes the Log-Derivative trick into an explicit sum of multiple Log-Derivative tricks, one for each of the categorical random variables present in the multivariate distribution. Next, we use the CatLog-Derivative trick to define a novel gradient estimator that exploits the structure of a probability distribution by following the variable ordering of the distribution's factorisation and effectively Rao-Blackwellises [4] REINFORCE.

**Definition 3.2** (The SCateR gradient estimator). We define the Structured Categorical REINFORCE (SCateR) estimator via the expression

$$\partial_\lambda \mathbb{E}_{\mathbf{X} \sim p(\mathbf{X})}[f(\mathbf{X})] \approx \sum_{d=1}^{D} \sum_{x_\delta \in \Omega(X_d)} \frac{1}{N} \sum_{n_d=1}^{N} \partial_\lambda p(x_\delta \mid \mathbf{x}_{<d}^{(n_d)}) f(\mathbf{x}_{<d}^{(n_d)}, x_\delta, \mathbf{x}_{>d}^{(n_d)}), \quad (3.5)$$

where the sample $\mathbf{x}_{>d}^{(n_d)}$ is drawn while conditioning on $x_\delta$ and $\mathbf{x}_{<d}^{(n_d)}$. The subscript on $n_d$ indicates that *different samples* can be drawn for every $d$.

**Proposition 3.3.** The SCateR estimator Rao-Blackwellises REINFORCE.

*Proof.* We start from Equation (2.4) for the REINFORCE estimator, where we ignore the first term and factorize the probability distribution similar to Equation (3.3)

$$\partial_\lambda \mathbb{E}_{\mathbf{X} \sim p(\mathbf{X})}[f(\mathbf{X})] \approx \frac{1}{N} \sum_{n=1}^{N} f(\mathbf{x}^{(n)}) \partial_\lambda \log p(\mathbf{x}^{(n)}) \approx \sum_{d=1}^{D} \frac{1}{N} \sum_{n=1}^{N} f(\mathbf{x}^{(n)}) \partial_\lambda \log p(x_d^{(n)}). \quad (3.6)$$

For notational conciseness, we drop the subscript on $n_d$ and simply use $n$ to identify single samples. Now we compare Equation (3.5) and Equation (3.6) term-wise for $N \to \infty$

$$\sum_{n=1}^{N} \mathbb{E}_{X_d \sim p(X_d | \mathbf{x}_{<d}^{(n)})} \left[ f(\mathbf{x}_{<d}^{(n)}, X_d, \mathbf{x}_{>d}^{(n)}) \partial_\lambda \log p(X_d | \mathbf{x}_{<d}^{(n)}) \right] = \sum_{n=1}^{N} f(\mathbf{x}_{<d}^{(n)}, x_d^{(n)}, \mathbf{x}_{>d}^{(n)}) \partial_\lambda \log p(x_d^{(n)}).$$

In the equation above, we see that SCateR takes the expected value for $X_d$ (left-hand side) and computes it exactly using an explicit sum over the space $\Omega(X_d)$, whereas REINFORCE (right-hand side) uses sampled values. This means, in turn, that the left-hand side is a Rao-Blackwellised version of the right-hand side. Doing this for every $d$ gives us a Rao-Blackwellised version for REINFORCE, i.e., the SCateR estimator. $\qquad\square$

**Corollary 3.4** (Bias and Variance)**.** The SCateR estimator is unbiased and its variance is upper-bounded by the variance of REINFORCE.

*Proof.* This follows immediately from Proposition 3.3, the law of total expectation and the law of total variance [3, 29]. $\qquad\square$

**Computational complexity.** Consider Equation (3.5) and observe that none of the random variables has a sample space larger than $K = \max_d(|\Omega(X_d)|)$. Computing our gradient estimate requires performing three nested sums with lower bound 1 and upper bounds equal to $D$, $K$ and $N$, respectively. These summations result in a time complexity of $\mathcal{O}(D \cdot K \cdot N)$. Leveraging the parallel implementation of prefix sums on GPUs [19], a time complexity of $\mathcal{O}(\log D + \log K + \log N)$ can be obtained, which allows for the deployment of SCateR in modern deep architectures.

**Function evaluations.** The main limitation of SCateR is the number of function evaluations it requires to provide its estimates. Indeed, each term in the three nested sums of Equation (3.5) involves a different function evaluation, meaning an overall number of at most $D \cdot \max_d(|\Omega(X_d)|) \cdot N$ function evaluations is necessary for every estimate. Fortunately, these evaluations can often be parallelised in modern deep architectures, leading to a positive trade-off between function evaluations and performance in our experiments (Section 6).

**Connection to Local Expectation Gradients (LEG).** LEG [36] was proposed as a gradient estimator tailored for variational proposal distributions. Similar to the CatLog-Derivative trick, it also singles out variables from a target expectation by summing over the variables' domains. However, it does so one variable at a time by depending on the Markov blanket of those variables. This dependency has the disadvantage of inducing a weighted expression, as variables downstream of a singled-out variable $X_i$ are sampled independently of the values assigned to $X_i$. In this sense, LEG can intuitively be related to Gibbs sampling where a variable is marginalised given possible assignments to its Markov blanket instead of being sampled.

In contrast, the CatLog-Derivative trick is applicable to any probability distribution with a known factorisation, even in the case where different factors share parameters. It sums out variables by following the variable ordering of this factorisation rather than depending on the Markov blanket. Following the topology induced by the distribution's factorisation naturally avoids the disconnect between summed-out and downstream variables for which LEG required a weighting term and makes the CatLog-Derivative trick more related to ancestral sampling. Moreover, computing these LEG weights adds additional computational complexity. A more technical and formal comparison between LEG and the CatLog-Derivative trick can be found in the appendix.

## 4 The IndeCateR Gradient Estimator

Using the CatLog-Derivative trick derived in the previous section we are now going to study a prominent special case of multivariate categorical distributions. That is, we will assume that our probability distribution admits the independent factorisation $p(\mathbf{X}) = \prod_{d=1}^{D} p_d(X_d)$. Note that all $D$ different distributions still depend on the same set of learnable parameters $\Lambda$. Furthermore, we subscript the individual distributions $p_d$ as they can no longer be distinguished by their conditioning sets. Plugging in this factorisation into Theorem 3.1 gives us the *Independent Categorical REINFORCE* estimator, or *IndeCateR* for short.

**Proposition 4.1** (IndeCateR). Let $p(\mathbf{X})$ be a multivariate categorical probability distribution that depends on a set of parameters $\Lambda$ and factorises as $p(\mathbf{X}) = \prod_{d=1}^{D} p_d(X_d)$, then the gradient of a finite expectation $\mathbb{E}_{\mathbf{X} \sim p(\mathbf{X})} [f(\mathbf{X})]$ can be estimated with

$$\sum_{d=1}^{D} \sum_{x_\delta \in \Omega(X_d)} \partial_\lambda p_d(x_\delta) \frac{1}{N} \sum_{n_d=1}^{N} f(\mathbf{x}_{\neq d}^{(n_d)}, x_\delta), \tag{4.1}$$

where $\mathbf{x}_{\neq d}^{(n_d)}$ are samples drawn from $p(\mathbf{X}_{\neq d})$.

*Proof.* We start by looking at the expression in Equation (3.1). Using the fact that we have a set of independent random variables, we can simplify $p(x_\delta \mid \mathbf{X}_{<d})$ to $p_d(x_\delta)$. As a result, the gradient of the expected value can be rewritten as

$$\partial_\lambda \mathbb{E}_{\mathbf{X} \sim p(\mathbf{X})} [f(\mathbf{X})] = \sum_{d=1}^{D} \sum_{x_\delta \in \Omega(X_d)} \mathbb{E}_{\mathbf{X}_{<d} \sim p(\mathbf{X}_{<d})} \left[ \partial_\lambda p_d(x_\delta) \mathbb{E}_{\mathbf{X}_{>d} \sim p(\mathbf{X}_{>d})} [f(\mathbf{X}_{\neq d}, x_\delta)] \right] \tag{4.2}$$

$$= \sum_{d=1}^{D} \sum_{x_\delta \in \Omega(X_d)} \partial_\lambda p_d(x_\delta) \mathbb{E}_{\mathbf{X}_{\neq d} \sim p(\mathbf{X}_{\neq d})} [f(\mathbf{X}_{\neq d}, x_\delta)] \tag{4.3}$$

Drawing $N$ samples for the $D - 1$ independent random variables $\mathbf{X}_{\neq d}$ and for each term in the sum over $d$ then gives us the estimate stated in the proposition. $\square$

> **Example 4.2** (Independent Factorisation). Let us consider a multivariate distribution involving three 3-ary independent categorical random variables. Concretely, this gives us
>
> $$p(X_1, X_2, X_3) = p_1(X_1)p_2(X_2)p_3(X_3), \tag{4.4}$$
>
> where $X_1$, $X_2$ and $X_3$ can take values from the set $\Omega(X) = \{1, 2, 3\}$. Taking this specific distribution and plugging it into Equation (4.1) for the IndeCateR gradient estimator now gives us
>
> $$\sum_{d=1}^{3} \sum_{x_\delta \in \{1,2,3\}} \partial_\lambda p_d(x_\delta) \frac{1}{N} \sum_{n_d=1}^{N} f(\mathbf{x}_{\neq i}^{(n_d)}, x_\delta). \tag{4.5}$$
>
> In order to understand the difference between the Log-Derivative trick and the CatLog-Derivative trick, we are going to to look at the term for $d = 2$ and consider the single-sample estimate
>
> $$\partial_\lambda p_2(1)f(x_1, 1, x_3) + \partial_\lambda p_2(2)f(x_1, 2, x_3) + \partial_\lambda p_2(3)f(x_1, 3, x_3), \tag{4.6}$$
>
> where $x_1$ and $x_3$ are sampled values for the random variables $X_1$ and $X_3$. These samples could be different ones for $d \neq 2$. The corresponding single sample estimate using REINFORCE instead of IndeCateR would be $\partial_\lambda p_2(x_2)f(x_1, x_2, x_3)$.
>
> We see that for REINFORCE we sample all the variables whereas for IndeCateR we perform the explicit sum for each of the random variables in turn and only sample the remaining variables.

In Section 6, we demonstrate that performing the explicit sum, as shown in Example 4.2, leads to practical low-variance gradient estimates. Note how, in the case of $D = 1$, Equation (4.1) reduces to the exact gradient. With this in mind, we can interpret IndeCateR as computing exact gradients for each single random variable $X_d$ with respect to an approximation of the function $\mathbb{E}_{\mathbf{X}_{\neq d} \sim p(\mathbf{X}_{\neq d})} [f(\mathbf{X}_{\neq d}, X_d)]$.

**LEG in the independent case.** The gap between the LEG and CatLog-Derivative trick gradient estimates grows smaller in the case the underlying distributions factorises into independent factors. The additional weighting function for LEG collapses to 1, resulting in similar expressions. However, two main differences remain. First, LEG does not support factors with shared parameters, although this problem can be mitigated by applying the chain rule. Second, and more importantly, LEG does not allow for different samples to be drawn for different variables. While this difference seems small, we will provide clear evidence that it can have a significant impact on performance (Section 6.4).

## 5 Related Work

Apart from LEG, another closely related work is the RAM estimator [38]. RAM starts by reparametrising the probability distribution using the Gumbel-Max trick followed by a marginalisation. We show in Section 3 that this reparametrisation step is unnecessary, allowing SCateR to explicitly retain the conditional dependency structure. Here we also see the different objectives of our work and Tokui and Sato's. While they proposed RAM in order to theoretically study the quality of control variate techniques, we are interested in building practical estimators that can exploit the structure of specific distributions.

Moreover, just like LEG, Tokui and Sato [38] did not study the setting of a shared parameter space $\Lambda$ between distributions, although this being the most common setting in modern deep-discrete and neural-symbolic architectures. Hence, an experimental evaluation for this rather common setting is missing in Tokui and Sato [38]. By providing such an evaluation, we show that efficiently implemented Rao-Blackwellised gradient estimators for categorical random variables are a viable option in practice when compared to variance reduction schemes based on control variates.

Such variance reduction methods for REINFORCE aim to reduce the variance by subtracting a mean-zero term, called the baseline, from the estimate [2, 28]. Progress in this area is mainly driven by multi-sample, i.e., sample-dependent, baselines [11, 12, 13, 25, 30, 37, 39] and leave-one-out baselines [17, 18, 26, 31, 34, 35]. Variance can further be reduced by coupling multiple samples and exploiting their dependencies [6, 7, 8, 9, 43, 44, 45]. These latter methods usually introduce bias, which has to be resolved by, for instance, an importance weighing step [8].

A general drawback of baseline variance reduction methods is that they often involve a certain amount of computational overhead, usually in the form of learning optimal parameters or computing statistics. This computational overhead can be justified by assuming that the function in the expectation is costly to evaluate. In such cases, it might pay off to perform extra computations if that means the expensive function is evaluated fewer times.

Another popular, yet diametrically opposite, approach to low-variance gradient estimates for categorical random variables is the concrete distribution [22]. The concrete distribution is a continuous relaxation of the categorical distribution using the Gumbel-Softmax trick [16]. Its main drawback is that it results in biased gradient estimates. Even though this bias can be controlled by a temperature parameter, the tuning of this parameter is highly non-trivial in practice.

## 6 Experiments

In this section we evaluate the performance of IndeCateR on a set of standard benchmarks from the literature. Firstly, two synthetic experiments (Section 6.1 and Section 6.2) will be discussed. In Section 6.3, a discrete variational auto-encoder (DVAE) experiment is optimised on three standard datasets. Lastly, we study the behaviour of IndeCateR on a standard problem taken from the neural-symbolic literature (Section 6.4).

We mainly compare IndeCateR to REINFORCE leave-one-out (RLOO) [17, 34] and to the Gumbel-Softmax trick. The former is a strong representative of methods reducing variance for REINFORCE while the latter is a popular and widely adopted choice in deep-discrete architectures [15].

We implemented IndeCateR and RLOO in TensorFlow [1], from where we also took the implementation of the concrete distribution.[1] Furthermore, all gradient estimators were JIT compiled. The different methods were benchmarked using discrete GPUs and every method had access to the same computational resources. A detailed account of the experimental setup, including hyperparameters and hardware, is given in the appendix.

In the analysis of the experiments we are interested in four main quantities: 1) the time it takes to perform the gradient estimation, 2) the number of drawn samples, 3) the number of function evaluations and 4) the value of a performance metric such as the ELBO. The number of function calls matters in as far that they might be expensive. Note, however, that evaluating the function $f$ is comparatively cheap and trivially parallelisable in the standard benchmarks.

---

[1]Our code is publicly available at `https://github.com/ML-KULeuven/catlog`.

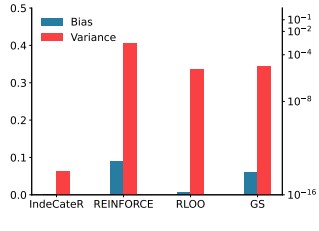
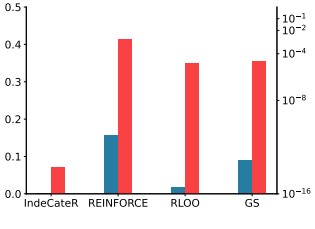
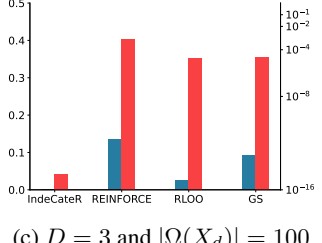

(a) $D = 12$ and $|\Omega(X_d)| = 3$     (b) $D = 6$ and $|\Omega(X_d)| = 10$     (c) $D = 3$ and $|\Omega(X_d)| = 100$

Figure 1: We report the empirical bias and variance for the different estimators and distributions in comparison to the exact gradient. The left y-axis in each plot indicates the bias while the right one shows the variance. Bias and variance results are averages taken over 1000 runs.

## 6.1 Synthetic: Exact Gradient Comparison

For small enough problems the exact gradient for multivariate categorical random variables can be computed via explicit enumeration. Inspired by Niepert et al. [27], we compare the estimates of

$$\partial_\theta \mathbb{E}_{\mathbf{X} \sim p(\mathbf{X})} \left[ \sum_{d=1}^{D} |X_d - b_d| \right] \tag{6.1}$$

to its exact value. Here, $\theta$ are the logits that directly parametrise a categorical distribution $p(\mathbf{X}) = \prod_{d=1}^{D} p(X_d)$ and $b_d$ denotes an arbitrarily chosen element of $\Omega(X_d)$. We compare the gradient estimates from IndeCateR, REINFORCE, RLOO, and Gumbel-Softmax (GS) by varying the number of distributions $D$ and their cardinality.

In Figure 1 we show the empirical bias and variance for the different estimators. Each estimator was given 1000 samples, while IndeCateR was only given a single one. Hence, IndeCateR has the fewest function evaluations as $D \cdot K$ is smaller than 1000 for each configuration. IndeCateR offers gradient estimates close to the exact ones with orders of magnitude lower variance for all three settings. RLOO exhibits the smallest difference in bias, yet it can not compete in terms of variance. Furthermore, the computation times were of the same order of magnitude for all methods. This is in stark contrast to the estimator presented by Tokui and Sato [38], where a two-fold increase in computation time of RAM with respect to REINFORCE is reported.

## 6.2 Synthetic: Optimisation

We now study an optimisation setting [37], where the goal is to maximise the expected value

$$\mathbb{E}_{\mathbf{X} \sim p(\mathbf{X})} \left[ \frac{1}{D} \sum_{i=1}^{D} (X_i - 0.499)^2 \right], \tag{6.2}$$

and $p(\mathbf{X})$ factorizes into $D$ independent binary random variables. The true maximum is given by $p(X_d = 1) = 1$ for all $d$. This task is challenging because of the small impact of the individual values for each $X_d$ on the expected value for higher values of $D$. We set $D = 200$ and report the results in Figure 2, where we compare IndeCateR to RLOO and Gumbel-SoftMax.

In Figure 2 and subsequent figures we use the notation RLOO-F and RLOO-S, which we define as follows. If IndeCateR takes $N$ samples, then it performs $D \cdot K \cdot N$ function evaluations with $K = \max_d |\Omega(X_d)|$. As such, we define RLOO-S as drawing the same number of samples as IndeCateR, which translates to $N$ function evaluations. For RLOO-F we match the number of function evaluations, which means that it takes $D \cdot K \cdot N$ samples. We give an analogous meaning to GS-S and GS-F for the Gumbel-SoftMax gradient estimator.

IndeCateR distinguishes itself by having both the lowest variance and quickest convergence across all methods, even compared to RLOO-F. Additionally, the time to compute all gradient estimates does not differ significantly for the different methods and leads to the same conclusions. It is striking to see that the Gumbel-Softmax estimator struggles in this task, which is likely due to its bias in combination with the insensitive loss function.

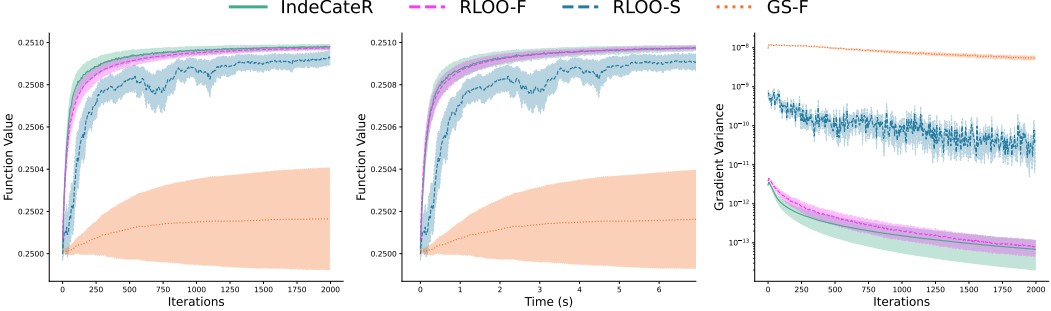

Figure 2: We plot the function value for different estimators against iterations (left) and time (middle). On the right, we plot the variance of the gradients against iterations. Statistics were obtained by taking the average and standard error over 10 runs. IndeCateR and RLOO-S both use 2 samples, while RLOO-F and Gumbel-Softmax (GS) use $800$ samples. The number of function evaluations is equal for IndeCateR, RLOO-F, and GS-F. We performed a hyperparameter search for the learning rate and the temperature of GS-F. Parameters were optimised using RMSProp.

## 6.3 Discrete Variational Auto-Encoder

As a third experiment we analyse the ELBO optimisation behaviour of a discrete variational auto-encoder (DVAE) [32]. We optimise the DVAE on the three main datasets from the literature, being MNIST [21], F-MNIST [42] and Omniglot [20]. The encoder component of the network has two dense hidden layers of sizes 384 and 256 ending in a latent 200-dimensional Bernoulli variable. The decoder takes samples from this variable followed by layers of size 256, 384 and 784. IndeCateR again uses two samples, hence we can compare to the same configurations of RLOO and Gumbel-Softmax as in Section 6.2. That is, equal samples (GS-S and RLOO-S) and equal function evaluations (GS-F and RLOO-F).

As evaluation metrics, we show the negated training and test set ELBO in combination with the variance of the gradients throughout training. We opted to report all metrics in terms of computation time (Figure 3), but similar results in terms of iterations are given in the appendix.

A first observation is that IndeCateR performs remarkably well in terms of convergence speed as it beats all other methods on all datasets in terms of training ELBO. However, we can observe a disadvantage of the quick convergence in terms of generalisation performance when looking at the test set ELBO. RLOO-F and IndeCateR both exhibit overfitting on the training data for MNIST and F-MNIST, resulting in an overall higher negative test set ELBO compared to the other methods. We speculate that the relaxation for the Gumbel-Softmax or the higher variance [41] of RLOO-S act as a regulariser for the network. As one would ideally like to separate discrete sampling layers from network regularisation, we see this overfitting as a feature and not a drawback of IndeCateR.

IndeCateR additionally is competitive in terms of gradient variance, especially considering the number of samples that are drawn. Across all datasets, the gradient variance of IndeCateR is comparable to that of RLOO-F, even though IndeCateR takes only 2 samples compared to the 800 of RLOO-F. The only method with consistently lower gradient variance is the Gumbel-Softmax trick, but only for equal function evaluations. However, in that case, the Gumbel-Softmax trick takes $\approx 1.6$ times longer than IndeCateR to compute its gradient estimates.

## 6.4 Neural-Symbolic Optimisation

A standard experiment in the neural-symbolic literature is the addition of MNIST digits [23]. We will examine an alternative version of this experiment, that is, given a set of $D$ MNIST digits, predict the sum of those digits. During learning, the only supervision provided is the sum and not any direct label of the digits. The difficulty of the problem scales exponentially, as there are $10^D$ states in the sample space. Note that there are only $10D + 1$ possible labels, resulting in only very sparse supervision.

In the field of neural-symbolic AI such problems are either solved exactly [23] or by simplifying the underlying combinatorial structure [14, 24]. Exact methods scale very poorly while simplifying the

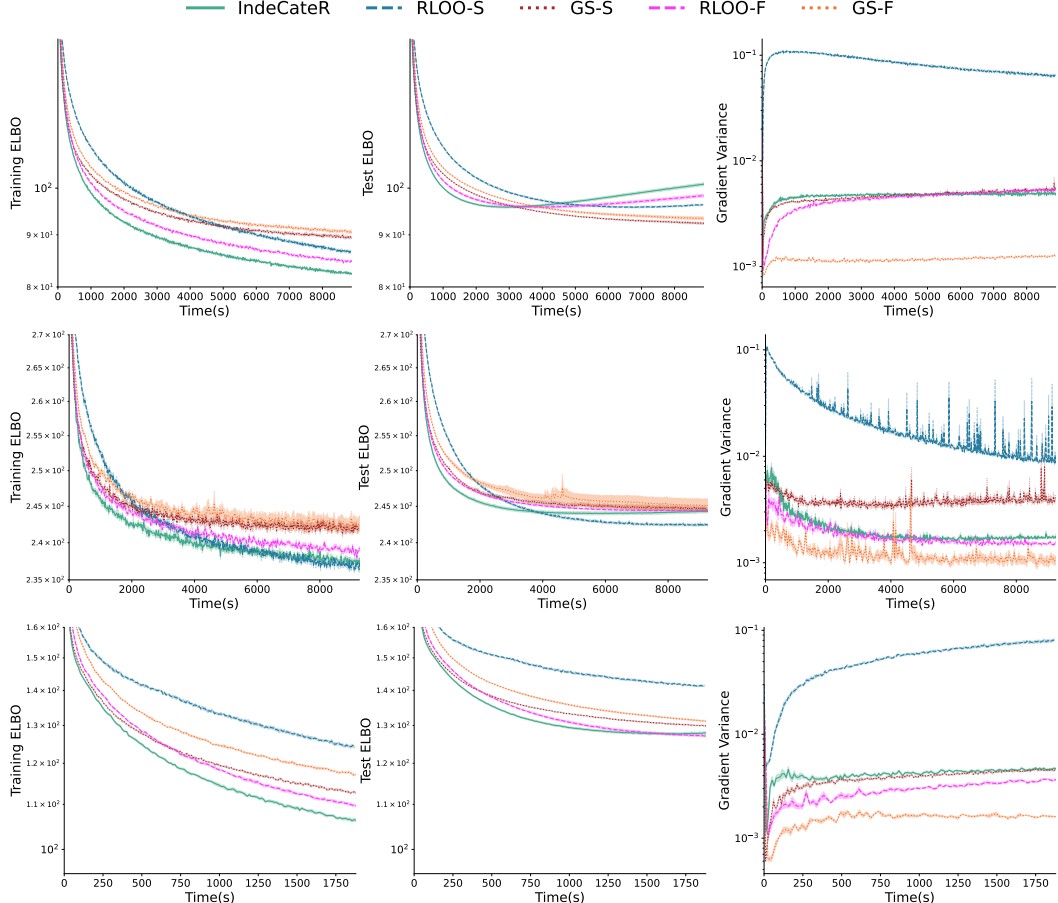

Figure 3: The top row shows negated training ELBO, negated test ELBO and gradient variance for the DVAE on MNIST, each plotted against time. Rows 2 and 3 show the same plots for F-MNIST (middle) and Omniglot (bottom).

combinatorics introduces problematic biases. In contrast, we will study neural-symbolic inference and learning using sampling and unbiased gradient estimators. The overall architecture is as follows, imitating inference in a general neural-symbolic system. Each of the D different MNIST images is passed through a neural classifier, which gives probabilities for each class. These probabilities are used to sample a number between $0$ and $9$ for each image. The numbers are summed up and compared to the label using a binary cross-entropy loss, as logical supervision is either true or false.

The hard part of this problem is the sparse supervision. In neural-symbolic AI, this sparsity is resolved by relying on classic search algorithms. However, these are not amenable to be run on parallel computing hardware, such as GPUs. In turn, training has to either occur entirely on the CPU or a hefty time penalty has to be paid for transferring data from and to the GPU.

Using IndeCateR in a neural-symbolic setting we achieve two things. On the one hand, we use sampling as a stochastic yet unbiased search, replacing the usual symbolic search. On the other, we render this stochastic search differentiable by estimating gradients instead of performing the costly exact computation. To make this work, we exploit a feature of IndeCateR that we have ignored so far.

In Equation (4.1) we can draw different samples for each of the $D$ different terms in the sum $\sum_{d=1}^{D}$ corresponding to the $D$ different variables. While drawing new samples for every variable does increase the total number of samples, the number of function evaluations remains identical. We indicate the difference by explicitly mentioning which version of IndeCateR draws new samples per variable. Note that this modification of the estimator would not have been possible following the work of Tokui and Sato [38] or Titsias, Michalis and Lázaro-Gredilla, Miguel [36].

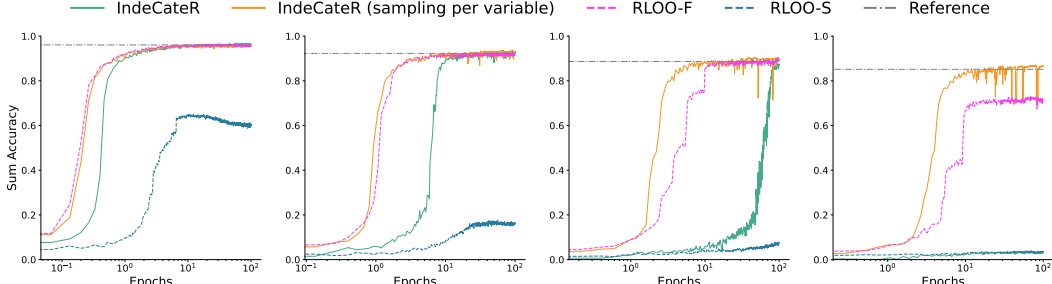

Figure 4: Test set accuracy of predicting the correct sum value versus number of epochs for the MNIST addition. From left to right, the plots show curves for the different estimators for $4$, $8$, $12$, and $16$ MNIST digits. We compare IndeCateR with and without sampling per variables to RLOO-F and RLOO-S. As the function $f$ has a zero derivative almost everywhere in this case, we can not compare to the Gumbel-Softmax trick. The dashed, grey line at the top of each plot represents a hypothetical sum-classifier that has access to a $99\%$ accurate MNIST digit classifier. The $x$-axis is in log-scale.

In Figure 4 we show the empirical results. We let IndeCateR without sampling per variable draw 10 samples such that IndeCateR with sampling per variables draws $D \cdot 10$ samples. The number of function evaluations of both methods stays the same at $D \cdot 10 \cdot 10$. In comparison, RLOO-F uses $D \cdot 10 \cdot 10$ samples and function evaluations. We see that IndeCateR is the only method capable of scaling and consistently solving the MNIST addition for 16 digits when taking new samples for every variable. In particular, these results show how the CatLog-Derivative trick estimates can significantly outperform the LEG and RAM estimates by drawing different samples for different variables, even in the independent case.

## 7  Conclusion

We derived a Rao-Blackwellisation scheme for the case of gradient estimation with multivariate categorical random variables using the CatLog-Derivative trick. This scheme led straightforwardly to a practical gradient estimator for independent categorical random variables – the IndeCateR estimator. Our experimental evaluation showed that IndeCateR produces faithful gradient estimates, especially when the number of samples is the main concern, and can be run efficiently on modern GPUs. Furthermore, IndeCateR constitutes the only hyperparameter-free, low-variance estimator for (conditionally) independent categorical random variables.

Three main axes can be identified for future work. First and foremost, we aim to analyse the more general case of a non-independently factorising probability distribution by applying SCateR to more structured problems, such as differentiable planning and reinforcement learning. Secondly, we plan to further study both IndeCateR and SCateR in the neural-symbolic setting and develop estimators for multivariate discrete-continuous distributions containing symbolic structures [5]. Finally, the combination of control variates with the CatLog-Derivative trick seems to be fertile ground for further improvements in variance reduction. Considering the Gumbel-Softmax Trick generally still exhibits lower variance for an equal amount of function evaluations, such a combination could put REINFORCE-based methods on equal footing with those based on the Gumbel-Softmax Trick.

## Acknowledgements

This research received funding from the Flemish Government (AI Research Program), from the Flanders Research Foundation (FWO) under project G097720N and under EOS project No. 30992574, from the KU Leuven Research Fund (C14/18/062) and TAILOR, a project from the EU Horizon 2020 research and innovation programme under GA No. 952215. It is also supported by the Wallenberg AI, Autonomous Systems and Software Program (WASP) funded by the Knut and Alice Wallenberg-Foundation.

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
