# OpenReview forum: "Differentiable Sampling of Categorical Distributions Using the CatLog-Derivative Trick"
_NeurIPS.cc/2023/Conference — NeurIPS 2023 poster_

### Official Review · Reviewer_8h59 · 2023-06-11

**Soundness:** 3 good
**Presentation:** 3 good
**Contribution:** 3 good
**Rating:** 7
**Confidence:** 4

**Summary:**

This paper proposes a variance-reduced statistic for $\mathbb{E}_{x \sim p(x)} [f(X)]$, which could be the gradient of a loss for example. The main idea (known as Rao-Blackwellization) is simple to understand and can be summarized thusly: the quantity can be rewritten to distinguish any dimension $d$:

$E_{(x_d, x_{\neq d}) \sim p(x_{d} | x_{\neq d}) p(x_{\neq d})} [f(X)]$ = $E_{x_{\neq d} \sim p(x_{\neq d})} E_{x_{d} \sim p(x_{d} | x_{\neq d})} [f(X)]$

Notationally, the difference in both sides is how $x_d$ is drawn. On the left hand side, $x_d$ is drawn jointly with the other dimensions on the right side: this means a one-to-one ratio of $x_d$ to $x_{\neq d}$. On the right hand side, $x_d$ is drawn separately from the other dimensions: this allows for a many-to-one ratio of $x_d$ to $x_{\neq d}$. In fact, when $x_d$ has finite outcomes, all of them can be used and the inner expectation (right hand side) can be computed exactly. The extra computation from the "many-to-one" sampling on the right hand side, is the price to pay for the variance-reduction that comes from using (averaging over) more $x_d$. And that computational price is acceptable for a categorical variable.

The authors show that the variance reduction is effective for a variety of tasks, whether it is the actual estimation of  $\mathbb{E}_{x \sim p(x)} [f(X)]$, or other tasks (e.g. optimization) that use the statistic for some other purpose such as stochastic optimization of a loss function.



**Strengths:**

The exposition is simple and clear, and the results are varied and convincing.

Figure 2 is also appreciated, as it empirically links the variance of the gradients (right panel) to the speed of the optimization (left and middle panels), which although it is common wisdom that has motivated certain algorithms (e.g. SVRG) is not a proven result for any optimizer.


**Weaknesses:**

No major weaknesses to indicate so far.

**Questions:**

It is not very clear from the text, that the statistic for  $\mathbb{E}_{x \sim p(x)} [f(X)]$ is sometimes the gradient of a loss (e.g. ELBO). For example, line 133 refers to a gradient without a loss without explicitly making the connection to $f(X)$. Could the authors clarify this in the text?


**Limitations:**

No specific concern on negative societal impact.

---

> ### Author Rebuttal · Authors · 2023-08-08
>
> We appreciate the concise explanation of our method and the positive assessment. Please, find below a remark about the formulas summarizing our method and the answer to your question.
>
> **Remark**
>
> Please, note that the CatLog-Derivative trick can be expressed in the following way (using your same notation):
> $E_{(x_{<d},x_d,x_{>d})\sim p(x_{<d},x_d,x_{>d})} \{ [f(X)] \}=E_{x_{<d}\sim p(x_{<d})} E_{x_d\sim p(x_d\mid X_{<d})} E_{x_{>d}\sim p(x_{>d}\mid x_{<d},x_d)} \{ [f(X)] \}$
>
> This emphasizes the relation to ancestral sampling, thus differentiating from other approaches like the local expectation gradients [1]
>
> [1] Titsias, M.K., & Lázaro-Gredilla, M. (2015). Local expectation gradients for black box variational inference. NeurIPS.
>
>
> **Question**
>
> We have added the following additional details to the appendix to clarify our optimisation setup for both the DVAE and neural-symbolic experiment.
>
> 1. Clarified that the loss function is the ELBO, which is itself an expected value. Additionally, the predicted probabilities used for the computation of the ELBO are clarified as well (Appendix A2, Modelling, end of paragraph).
>
> 2. We explicitly added the target loss function $-\log P(\sum_{i = 1}^D d_i = s)$ for the neural-symbolic optimisation to the appendix and expressed it using an expected value (Appendix A.3, Modelling, end of paragraph + Eq. A.1).
>
> # Final Comments
>
> Please, let us know if there is anything you want to discuss.

---

> > ### Comment · Reviewer_8h59 · 2023-08-14
> > **Response from Reviewer 8h59**
> >
> > I thank the authors for their response and appreciate the discussion comparing CatLog to LEG, which I was not previously aware of. I am still following that discussion.
> >
> > So long as the differences between CatLog and LEG are clearly explained and LEG is properly referenced, I see no reason to lower my score and there still seems to be a novel contribution with empirical results to support it.

---

### Official Review · Reviewer_szeH · 2023-07-05

**Soundness:** 2 fair
**Presentation:** 2 fair
**Contribution:** 1 poor
**Rating:** 3
**Confidence:** 4

**Summary:**

The paper discusses improved gradient estimation through multivariate categorial probability distributions. A special case of independent random variables is discussed more thoroughly and experiments are conducted only for this case. To summarize for independent variables i.e. $p(X) = \prod_{i \in [N]} p(X_i)$ following is my understanding:

1. Assume for applying REINFORCE trick one would use $N$ many $d-$dimensional samples $x^{(n)}$ drawn from $p(X)$.
2. This paper proposes the following:
* Create $N$ many $d-$dimensional samples similar to 1.
* For each variable $i$ in $[D]$ enumerate all possible values (assuming $K$ many) of variable $i$ and replace $X_i$ in each  $x^{(n)}$. Thus creating $K$-times additional samples for each variable.


**Strengths:**

1. The idea of looking 'inside' a joint probability distribution instead of treating it as a black-box is interesting. For independently distributed variables the paper makes use of this factorization.
2. Experiments (Fig. 2, 3, 4) indicate that proposed method _only sometimes_ produces better gradient estimates than REINFORCE on equal footing.

**Weaknesses:**

3. It would have been great to have an illustration of the case discussed in Example 4.2 to see things (and symbols) pictorially. Preferably at the start of paper.
4. As discussed in line 139, the computational complexity of this approach is $\mathcal{O}(D \cdot K \cdot N)$. This looks much expensive than what REINFORCE has which is just $\mathcal{O}(N)$. Although for a fair comparison in experiments the authors do allow REINFORCE to have more samples which is termed as RLOO-F. Therefore in following I will only compare with such baselines.\
a. First two rows in Figure 2 has only positive results of the proposed approach although only with slight improvements over RLOO-F.  \
b. In last row of Figure 2, GS-F is better (gumbel softmax on equal footing) with RLOO-F not so far behind. \
c. In Figure 4 the proposed method does far worse than RLOO-F! Unless it is allowed to sample $x^{(n)}$ individually for each $d$ in $[D]$. This is a serious shortcoming.
5. Figure 1 does not contain RLOO-F, why?
6. The discussed benchmarks are too small and old. It would have been better to discuss a more practical and large-scale problem to solve.
7. In Example 4.3 it will be good to not have $K = D = 3$ as it makes it more difficult to parse. Possibly make $K=2$ (i.e., binary variables).
8. Can there by any benefits of proposed approach on cases where variables are not independent? Possibly experiments can be added to put estimator in eq. 3.5 to test in a future submission.

**Questions:**

9. How does time complexity change due to parallelization in line 140?
10. Line 138 end: respectively w.r.t. what?
11. What about training accuracy for Figure 4?

**Limitations:**

12. Overall the method falls short of its promise. It needs much computational complexity to create additional samples and REINFORCE trick does sometimes even better than proposed method if tested on equal footing.

---

> ### Author Rebuttal · Authors · 2023-08-08
>
> We thank the reviewer for the time dedicated to review our paper. Please, find below a discussion about the mentioned weaknesses and the answers to the questions.
>
> # Discussion
>
> The improvements with respect to RLOO-F are statistically significant, as can be seen from the reported standard error bars shown in all graphs. We also want to emphasise that the improvements over RLOO-F seem to increase with increasing K (domain size of categorical distributions). This can be seen both from Figure 1 and 4. In the former, the bias for RLOO-F increases with increasing K. In the latter, IndeCateR-D is the only method capable of providing consistent solutions to the MNIST addition problem. While regular IndeCateR does not beat RLOO-F in experiment 6.4, it has to be taken into account that RLOO-F also takes $10\cdot D \cdot N$ samples in contrast to just $10$ samples for IndeCateR. IndeCateR-D equalises both number of samples drawn and function evaluations and is as such the main competitor for RLOO-F. In the other experiments, due to the mainly binary domain of each variable, IndeCateR-D did not give meaningful improvements over IndeCateR. In the binary cases, IndeCateR takes fewer samples than RLOO-F and still manages to significantly outperform it.
>
> With respect to the last row of Figure 3, IndeCateR does lose out to GS-S. However, the Omniglot dataset was run on an older GPU (GTX 1080 Ti) in contrast to the MNIST and F-MNIST datasets. We have rerun Omniglot on the same GPU (RTX 3080 Ti) as MNIST and F-MNIST and can see that IndeCateR can make better use of the power of newer AI accelerators as it now again outperforms all other methods while all methods exploit parallelisation where possible. We have added this figure to the paper.
> In Figure 1, all methods were given 1000 samples while IndeCateR was only given 1. This choice was made to show that even when competitors are given more function evaluations and more samples, IndeCateR can still outperform them. In short, Figure 1 compares IndeCateR to an even stronger estimator than RLOO-F.
> The choice of benchmarks follows the general and expected experimental setup from the gradient estimation literature [1, 2]. We would argue that the DVAE experiment in particular presents a significant challenge for gradient estimators as it has relatively high dimensionality in conjunction with a neural optimisation component.
> Regarding benchmarks. Our primary focus was on the most expected benchmarks from the gradient estimation literature, which turned out to all be cases of fully factorising distributions [1, 2]. Does the reviewer have any other particular benchmark in mind?
>
> [1] Jang, Eric, Shixiang Gu, and Ben Poole. "Categorical Reparameterization with Gumbel-Softmax." ICLR (2016).
>
> [2] Richter, Lorenz, et al. "VarGrad: a low-variance gradient estimator for variational inference." NeurIPS (2020).
>
> # Questions
>
> **How does time complexity change due to parallelisation in line 140?**
>
>
> Our summations can be cast as special cases of the prefix sum [3], which has a parallel implementation with complexity $O(\log N)$ [4] when summing over $N$ terms.
>
> [3] https://en.wikipedia.org/wiki/Prefix_sum
>
>
> [4] Ladner, Richard E., and Michael J. Fischer. "Parallel prefix computation." Journal of the ACM (1980).
>
> **Line 138 end: respectively w.r.t. what?**
>
>
> The respective upper bounds are for each of the three nested sums.
>
> **What about training accuracy for Figure 4?**
> We are interested in the generalisation performance of the classifiers, which is quantified by the test set accuracy. Training accuracy is generally not considered interesting for classification problems as it is sensitive to overfitting, hence we left it out for clarity of exposition.
>
> # Final Comments
>
> We hope that we have addressed all of your concerns. Please, let us know if there is anything else you want to discuss.

---

> > ### Comment · Reviewer_szeH · 2023-08-14
> >
> > **In the former, the bias for RLOO-F increases with increasing K:**
> > Figure 1 does not have RLOO-F, only RLOO.
> >
> > **IndeCateR-D is the only method capable of providing consistent solutions to the MNIST addition problem:**
> > Agreed (Figure 4).
> >
> > **RLOO-F takes more samples than IndeCateR:**
> > Both do use same number of function evaluations (Line 219). In RLOO-F the arguments for function evaluation are fully random while in IndeCateR they are hand-designed. Why is it important then to signify that RLOO-F takes more 'samples'?
> >
> > **Figure 3, slower GPU:**
> > One could show epochs in x-axis instead of time, as also was done in Figure 4? While we are at it, why the inconsistency in x-axes of Figure 3 and 4?
> >
> > Looking forward to your response.

---

> > > ### Author Response · Authors · 2023-08-16
> > > **Addressing additional questions**
> > >
> > > Thank you for the additional questions and interest. We hope the following answers your concerns to a satisfactory degree.
> > >
> > > **Figure 1 does not have RLOO-F, only RLOO.**
> > >
> > > Indeed RLOO-F is a typo in our answer. Figure 1 is supposed to just have RLOO with 1000 function evaluations. This has more function evaluations than RLOO-F would require for this experiment and hence shows we are able to outperform even stronger baselines in some cases.
> > >
> > > **Why is it important then to signify that RLOO-F takes more 'samples'?**
> > >
> > > Taking more samples introduces additional computational costs both because of the sampling itself and any downstream operations. This computational cost can lead to significant differences in performance per unit of time (RLOO-F vs IndeCateR).
> > >
> > > **Different x-axes**
> > >
> > > In Figure 3 we wanted to emphasize the computational efficiency of IndeCateR versus other methods. However a figure in function of iterations is also given in the appendix. There was no significant difference in computational time for Figure 4, just as was the case in Figure 2 where we do show both time and iterations on the x-axis. For clarity of exposition, we hence chose to only report epochs for Figure 4.
> > >
> > >
> > > The existing difference in computational time in Figure 3 is caused by the additional costs of backpropagating through the additional samples for RLOO-F. In other words, IndeCateR has similar computational requirements to RLOO-S, but performance closer to RLOO-F.

---

### Official Review · Reviewer_i3wk · 2023-07-06

**Soundness:** 3 good
**Presentation:** 2 fair
**Contribution:** 3 good
**Rating:** 7
**Confidence:** 3

**Summary:**

In this paper the authors provide an alternative to the log derivative trick, e.g. the REINFORCE estimator, for categorical distributions, which is unbiased and has provably lower variance than the REINFORCE estimator; the estimator is called the CatLog-Derivative trick. For the case D-dimensional multivariate categorical distributions, where each dimension is independent, the authors introduce a simplification of the CatLog-Derivative trick called IndeCateR.

**Strengths:**

The proposed idea is elegant and simple and the authors did a great job explaining it! Actually, I'm surprised they were able to fit the proofs inside the main text, which speaks to the conciseness of the paper. I also think the authors did a good job explaining the limitations of the base CatLog trick approach and the variety of experiments were great as well.

**Weaknesses:**

The major weakness is that there is no experiment demonstrating the performance of the CatLog trick. While emphasis is placed on the IndeCateR trick, it would be great the CatLog trick in action. As a suggestion, maybe a variational hidden markov model?

**Questions:**

- There are a couple of typos in the paper and sentences that aren't complete, i.e. beginning of line 177
- Figure 1 missing y labels, making it hard to parse without reading
- It would be great to have a section in the appendix doing a brief overview of alternative methods
- Figure 4 seems to have problems rendering. I suggest rasterizing the figure and putting it back in the appendix.
- In general, all line plots are hard to read. I suggest increasing the thickness of the lines.

**Limitations:**

Yes, they have addressed the limitations of the method.

---

> ### Author Rebuttal · Authors · 2023-08-08
>
> Thank you for the enthusiasm and the constructive suggestions. Please find below a comment about the weaknesses and the answers to your questions.
>
> # Comment
>
> We acknowledge that we have primarily devoted attention to IndeCateR, instead of the more general CateR. Our primary focus was on the most expected benchmarks from the gradient estimation literature [1, 2], which turned out to all be cases of fully factorising distributions. We really like the suggestion of a variational HMM as a testbed for CateR and look forward to adding this to an extension of this work!
>
> # Questions
>
> **Typos**
>
> Thanks for spotting them ! We have made the following modifications:
> - Line 41: “reparametrisation for categorical distributions, which we discuss further in the related work (Section 5)”
> - Line 176-177: “Firstly, two synthetic experiments (Section 6.1 and Section 6.2) will be discussed.”
>
> **Figure 1**
>
> This was indeed not clear and labels were added to the figure.
>
> **Appendix about brief overview**
>
> The page limit did prevent us from providing a more extensive overview of possibly related work. We have taken this suggestion into account and added a more complete related work section to the appendix (Appendix B.1).
>
> **Figure 4 seems to have problems rendering**
>
> Thanks for the suggestion! Indeed, there was a problem with the rendering.
>
> **In general, all line plots are hard to read. I suggest increasing the thickness of the lines**
>
> Thanks for the suggestion to improve the readability of the figures. We have taken it into account and modified them accordingly.
>
> [1] Jang, Eric, Shixiang Gu, and Ben Poole. "Categorical Reparameterization with Gumbel-Softmax." ICLR (2016).
>
> [2] Richter, Lorenz, et al. "VarGrad: a low-variance gradient estimator for variational inference." NeurIPS (2020).

---

### Official Review · Reviewer_JDGY · 2023-07-13

**Soundness:** 3 good
**Presentation:** 4 excellent
**Contribution:** 1 poor
**Rating:** 4
**Confidence:** 5

**Summary:**

The authors derive a gradient estimator for discrete random variables that sums out one dimension while keeping a sample for the other dimensions fixed. Their estimator works for generally fully factorised distributions, and they derive a variant for fully independent distributions, with which they perform experiments on the discrete VAE and a neurosymbolic task.

**Strengths:**

The paper is very easy to follow and well-motivated. The estimator is quite interesting and competitive with the strong RLOO estimator. The experiments are useful to the community, and especially showing that likelihood-ratio-based methods work for neurosymbolic methods is a useful insight.


**Weaknesses:**

I'm afraid that the paper is very limited in novelty. The approach presented is almost the same as Local Expectation Gradients (LEG) [1], but LEG is not discussed in the paper. From my understanding, this is how they compare:

Similarities:
- The CatLog-Derivative Trick (Eq 3.5) is the LEG (Eq 9) under a fully factorised distribution / autoregressive model.
- The IndeCateR (Eq 4.1) is precisely the simplification of LEG for fully independent distributions in Eq 11
- LEG also connects the estimator to Rao-Blackwellization

Differences:
- LEG uses a 'pivot' sample and then sums over dimensions, while CatLog resamples for each dimension
- LEG (like RAM) did not study the shared parameter setting, although, in my opinion, this is a trivial extension by backpropagation

Therefore, I don't think the paper can claim a new trick, as (at best) it slightly modified an established work. That is not to say the results of this paper are not useful but need to be recontextualised, given that the method is not novel. Suggestions are a more thorough experimental setup or a focus on neurosymbolic tasks (which are somewhat understudied in the literature on discrete gradient estimation).

[1] Titsias, M.K., & Lázaro-Gredilla, M. (2015). Local expectation gradients for black box variational inference. Advances in neural information processing systems, 28.

EDIT: The author rebuttal showed that CatLog indeed has some novelty in its sampling mechanism, and I increased my score from 3 to 4. I still think this paper lacks novelty: While CatLog has some novelty, there are no experiments that use it. The only experiments are with IndeCateR, but that method is not novel (LEG and REM on independent distributions are the same as IndeCateR as the authors acknowledge in their rebuttal). IndeCateR-D has some novelty compared to LEG and REM in sampling, but this method is only tested in a single experiment and is not highlighted in the writing as the focus of the paper.

**Questions:**

- Experiments: Why not compare to RAM, as it is closest to the method?
- Figure 1: What are the 2 y-scales?
- Line 199: Do you assume one-hot encoded X_d?
- Line 235: You claim IndeCateR uses two samples, but I do not see how that works. There are 200 dimensions, so you will get 200 samples if you resample per dimension. Or do you set N=2, and use a pivot like in LEG?
- Did you evaluate the MNIST experiments on multiple runs? I do not see error bars here. These MNISTAdd experiments can have significant variance between runs, so I believe having at least 10 runs is necessary before claiming IndeCateR is the only method that can scale to 16 digits.
- Clarify that the MNIST experiments are different from the multi-digit MNISTAdd experiments discussed in [1] (it's not $100x_1 + 10x_2 + x_3+100x_4+10x_5+x_6$, but rather $\sum_i x_i$). These have quite different optimisation properties (the multi-digit MNISTAdd problem has $10^{2/D}-2$ labels rather than $10D+1$)

[1] Manhaeve, R., Dumančić, S., Kimmig, A., Demeester, T., & De Raedt, L. (2021). Neural probabilistic logic programming in DeepProbLog. Artificial Intelligence, 298, 103504.

Typos:
- Line 177: Unfinished sentence
- Line 207: RLOO-F mentioned before its introduction
- Line 219: Comma after samples
- Line 265: Comma instead of point

**Limitations:**

I would say the limitations are adequately addressed.

---

> ### Author Rebuttal · Authors · 2023-08-08
>
> We thank the reviewer for the dedicated time and for bringing the related work on local expectation gradients (LEG, NeurIPS 2015) [1] to our attention. We also thank them for their appreciation about the clarity, significance and quality of our work. Please, find below a discussion on the relation between CatLog and LEG and the answers to your questions.
>
> # Comparison between CatLog and LEG
>
> Thanks for pointing us to LEG! Indeed, LEG and IndeCateR are related. However, CatLog and LEG are two substantially different tricks/methods for the following reasons:
>
> 1. LEG does not make full use of the autoregressive parametrisation of the distribution, as it obtains samples (aka pivots) by first instantiating all variables and subsequently performing evaluation through the computation of a weighted average based on the Markov blanket. CatLog instead makes full use of the autoregressive parametrisation by interleaving sampling and evaluation. LEG is based on three distinct stages: sampling, evaluation and weighted averaging. In contrast, CatLog has only two intertwined stages based on sampling and evaluation. Another way/analogy to look at them is that LEG is inspired and related to Gibbs sampling, whereas CatLog is based on ancestral sampling. Please, refer to the detailed analysis of LEG in the attached PDF, which will be included in the Appendix of our paper and also the table underneath summarizing the main differences.
>
> 2. The two methods have different computational complexity. Indeed, computing the weighted average in LEG requires an additional $O(D)$ cost, which contributes to an overall computational complexity of $O(ND^2K)$. Therefore, the CatLog-Derivative trick makes a better use of the structure, resulting in improved efficiency over LEG, as scaling only linearly with the number of variables.
>
> 3. As the reviewer mentioned, the sampling per dimension introduced by CatLog is different from the pivot samples used by LEG. This theoretical difference also yields significant practical differences in performance, even in the case of an independently factorising distribution. This difference can be seen in Section 6.4. There, the ‘IndeCateR’ estimator corresponds to pivot samples (LEG), while ‘IndeCateR-D’ draws new samples per dimension (CatLog). IndeCateR-D is the only estimator that can consistently tackle the problem when increasing the dimensionality, showing that CatLog is both theoretically and practically different from LEG.
>  (RESAMPLING STUFF? SEE ALSO DOUBTS BELOW)…
>
> **Table summarizing the main differences**
>
> | Name  |  Trick | Computational Complexity | Relation to Sampling |
> |---|---|---|---|
> | LEG  | $\sum_{d=1}^D E_{(X_{<d},{\color{red}X_d'},X_{>d})\sim p(X_{<d},{\color{red}X_d'},X_{>d})} \{ E_{X_d\sim {\color{blue}p(X_d\mid X_{\neq d})}}[f(X)\partial_\lambda \log p(X_d\mid X_{<d})] \}$  | $O(ND^2K)$  | Gibbs sampling  |
> | CatLog  | $\sum_{d=1}^D E_{(X_{<d},{\color{red}X_d},X_{>d})\sim p(X_{<d},{\color{red}X_d},X_{>d})} \{ [f(X_{\neq d}, X_d)\partial_\lambda \log p(X_d\mid X_{<d})] \}$ |  $O(NDK)$ | Ancestral sampling |
>
> We made the following modifications to the text:
>
> The attached PDF containing the formal proof of the theoretical difference between CatLog and LEG is added to the Appendix (Appendix B.2).
> We highlight the difference between taking samples per dimension (CatLog) and not doing so (LEG) in Section 6.4 by further detailing the difference between IndeCateR and IndeCateR-D following our above answer.
>
> # Answers to Questions
>
> **Experiments: Why not compare to RAM, as it is closest to the method?**
>
> When not taking new samples per dimension, CatLog, LEG and RAM all collapse to the same estimate for an independently factorising distribution. In Section 6.4, we look at both cases with (IndeCateR-D) and without sampling per dimension (IndeCateR, RAM and LEG) per dimension and observe that drawing new samples essentially makes the difference between being able to solve the problem or not. As such, CatLog does provide a measurable improvement over RAM and LEG.
>
> **Figure 1: What are the 2 y-scales?**
>
> The left y-scale concerns the bias while the right one looks at variance. We have added these labels to the figure for clarity.
>
> **Line 199: Do you assume one-hot encoded X_d?**
>
> We do not use one-hot encoded vectors throughout the paper, only direct elements of the categorical domains.
>
> **Line 235: You claim IndeCateR uses two samples, but I do not see how that works. There are 200 dimensions, so you will get 200 samples if you resample per dimension. Or do you set N=2, and use a pivot like in LEG?**
>
> We use IndeCateR without drawing new samples every dimension in the binary DVAE experiment. In essence, we take 2 samples of the joint distribution and for dimension $d$, we use the components $\neq d$ to perform the estimation. This choice was made based on the empirical observation that drawing new samples per dimension in the case of binary random variables did not give a measurable improvement.
>
> **Did you evaluate the MNIST experiments on multiple runs? I do not see error bars here. These MNISTAdd experiments can have significant variance between runs, so I believe having at least 10 runs is necessary before claiming IndeCateR is the only method that can scale to 16 digits.**
>
> We did evaluate the MNIST experiment across 5 runs and error bars are reported in Figure 4. It does seem that, depending on the PDF viewer, these error bars do not render properly on certain zooming levels. Please try using a different PDF viewer to see the error bars that confirm IndeCateR-D consistently beats the competitors. We will also rasterize the image in the final version of the paper to avoid the potential visualisation issue.
>
> **Clarify that the MNIST experiments..**
>
> Thanks for the suggestion, we will add this clarification to the main paper.
>
> # Final Comments
>
> Please, let us know if your concerns have been addressed and if you have any further question, we would be happy to answer.

---

> > ### Comment · Reviewer_JDGY · 2023-08-10
> >
> > I thank the authors for their insightful rebuttal. I did not consider the difference between Gibbs sampling and ancestral sampling between LEG and CatLog, and indeed there is some novelty in CatLog. The complexity bounds are also useful. I will raise the score, but only to 4.
> >
> > I think the paper is currently missing a clear story, taking the concerns of the reviews into account. It is written as if introducing a significant new trick, but then shows it is a variation of LEG. There are probably benefits to ancestral sampling, but this is not shown as the authors only experiment with IndeCateR. Furthermore, IndeCateR _itself_ is not novel, as the authors acknowledge (it is equivalent to both REM and LEG). IndeCateR-D is slightly different, but is only introduced in the last experiment, and (according to the reviewers) did not lead to an improvement in DVAE.
> >
> > This could be a very strong paper, but from what I could review, I have trouble describing the core contribution. The paper could add eg experiments on autoregressive models to showcase the use of ancestral sampling, or highlight the differences between 'pivot' sampling and sampling per dimensions. (But these are of course just suggestions).
> >
> > Please let me know if there are any misconceptions
> >
> > EDIT: For some reason, I cannot modify my review right now. I will visit this page again later to do so.

---

### Official Review · Reviewer_Z21m · 2023-07-13

**Soundness:** 3 good
**Presentation:** 3 good
**Contribution:** 4 excellent
**Rating:** 7
**Confidence:** 4

**Summary:**

This paper develops a new efficient estimator for gradients of an expectation computed over a multivariate discrete random variable. The gradients are computed wrt the continuous parameters of the discrete distribution of interest.

Previous work consists of two major threads
- unbiased estimators, such as REINFORCE (aka the score function trick or log-derivative trick)
- biased estimators, such as relaxed Gumbel-softmax trick aka "concrete" random variables [refs 15,20]

To build on these, this paper presents an approach that is:
* formally unbiased (like REINFORCE, unlike GS)
* has low variance (unlike REINFORCE)
* no free hyperparameters (other than the number of Monte Carlo samples), unlike control variate extensions of REINFORCE

In Sec 3, the general approach, called CateR, is presented.
The big idea is to Rao-Blackwellize REINFORCE across each dimension of the multivariate discrete vector X being sampled. See Eq 3.1 for the estimator.

In Sec 4, a straightforward specialization of the estimator to a model that assumes each dimension is *independent*, caled IndeCateR, is presented. Example 4.2 gives nice intuition. This requires nested sums over the dimensions D, the K possible values of each variable, and N sampled values of remaining variables, with runtime O(DKN).

Experiments in Sec 6 assess:
* empirical bias/variance on synthetic problems (Fig 1)
* performance in optimization on a toy problem (Fig 2)
* performance in training of discrete VAEs (Fig 3)
* a "neuro-symbolic" model that tries to compute the sum of 10 MNIST digit images, by sample a predicted digit label (discrete value in 0-9) for each image then adding

Across the board, the experiments suggest the estimator is competitive in performance when computation cost is similar to its alternatives.

**Strengths:**


I found a lot to like about this paper.

+ Attacks a significant problem: many models need such a gradient estimator and current methods (like REINFORCE) are known to be difficult due to high variance
+ Elegant yet simple method: derivable from first principles and interpretable as Rao-Blackwellization
+ Comprehensive experiments on toy data help gain intuition about when/why the method works
+ Overall message and formal mathematics are both clearly communicated throughout
+ Effort to make costs fair for all methods (in terms of num func evaluations and/or num samples) is appreciated

Thanks to the authors for their hard work


**Weaknesses:**


Overall I don't think there are show-stopping weaknesses here. I'll list some issues below, but I'd overall rate these as definitely worth addressing but minor.

### Improve discussion of AI accelerators

The paper throughout refers to "modern AI accelerators" without much elaboration or citation, leaving unfamiliar readers in the dark. I think there could be many different kinds of hardware accelerator the authors are thinking of...

* What does it mean that this approach can be implemented on a modern AI accelerator? Would this be true of alternatives? like REINFORCE or GS?
* How would such accelerators improve the runtime of IndeCateR from O(DKN) to O(log D + log K + log N)? Aren't they just reducing constant factor runtime by moving computation from software to hardware?

### Claim that temperature hyperparameter of GS methods is difficult to tune needs elaboration

In line 172-173, GS / concrete methods are criticized because "tuning of this [temperature] hyperparameter" is "highly non-trivial in practice".

Can you provide some evidence for this claim?
I don't doubt that it *could* be sensitive, but I wonder if your experiments could reveal more about this. For example, in Fig 1 or Fig 2 you could show GS with tuned hyperparameter compared to GS with a reasonable default value.

### Is there recent work extending GS/concrete that's worth discussing/comparing?

The concrete distribution / GS methods [refs 15,20] were published about 6 years ago, in 2017.

Seems like the current related work discussion doesn't really touch on any progress that might have been made since 2017. I'm not aware off the top of my head of such work, but I plan to dig in more during discussion period to see if there's relevant work.

If so, definitely seems worth citing a few more papers. If not, perhaps highlighting the lack of further progress there is interesting.


### Presentation quality decent but needs refinement to catch isolated issues

At a few spots there are incomplete thoughts or awkward phrasings, such as
* line 41
* line 177

In further revision, please read carefully to catch such issues and spare a future reader any confusion.


### DVAE experiments need more details/elaboration

* Need to motivate why the DVAE is an interesting model
* Does the original DVAE recommend a specific gradient estimator? If so, what?
* For encoder arch, are there really 3 *hidden* layers, or just 2 hidden layers and one output layer?
* Why 2 samples for IndeCateR instead of just 1 or many more? How sensitive is this choice?
* Can you define the probabilistic model and the optimization problem? (perhaps in supp.). In particular, is your likelihood treating the pixels as unconstrained real floats, as floats in 0.0 - 1.0, or binary values?
* Can you clarify the optimization algorithm used (SGD? ADAM?)?
* Where methods compared using the same initial parameter values? Or similar sampling scheme for initialization?
* Is there any regularization applied? (e.g. L2 penalty on encoder/decoder parameters)



**Questions:**


### Q1: How did you compute bias and variance in Fig 1?

Presumably your estimated gradient is a vector, not a scalar.
Did you just add or average each component's bias/variance to get the scalar metrics reported here?

### Q2: Can you try to give insight into why IndeCateR has lowest variance in Fig 1 and 2, but GS has lower variance with DVAEs in Fig 3?

What is the key difference here?


### Q3: Why do we need sampling at all in Sec 6.4? And why frame as a binary classifier?

I'm fine with what's presented in Sec 6.4 as a proof of concept that your approach can work on this problem.

But is there an *advantage* to formaluating the problem by requiring each per-image predictor to produce a random sample, that is then summed? Why not just have each per-image classifier produce the expected value of the categorical distribution over digits 0-9, and sum that across images?

Also, why use a binary (right/wrong) signal to supervise? Why not use a more continuous error metric, since if the true sum is 33 but I predict 32 I'd probably prefer that vastly compared to predicting 0.



Minor questions

* Why isn't GS-S shown in Fig 2?

**Limitations:**

Could say more about limitations in Sec 7. There's at least 1 paragraph worth of room for it.

---

> ### Author Rebuttal · Authors · 2023-08-08
>
> Thanks for the detailed and thoughtful review and the extra effort to provide feedback aiming at improving the quality of the presentation and at sharpening some statements. Please, find below a discussion to your raised points and the answers to your questions.
>
> # Discussion about weaknesses
>
> **AI accelerators**
>
> We acknowledge that the statements are too vague. The main AI accelerators that we have in mind are GPUs. We are going to modify the text in the following way:
>
> - Line 44: "can be implemented efficiently by leveraging parallelisation on modern graphical processing units (GPUs)."
>
> - Line 139-141: "Leveraging the parallel implementation of prefix sums [1] on GPUs [2], the practical runtime can be reduced to $\mathcal{O}(\log D + \log K + \log N)$,...."
>
> - Line 271: "these are not amenable to be run on parallel computing hardware, such as GPUs."
>
> - Line 294: "modern GPUs."
>
> [1] https://en.wikipedia.org/wiki/Prefix_sum
>
> [2] Ladner, Richard E., and Michael J. Fischer. "Parallel prefix computation." Journal of the ACM (1980).
>
> **Tuning of hyperparameters of GS**
>
> Experimental support for this claim was not explicitly provided for simplicity of exposition. However, we did encounter the difficulties of temperature tuning during our experimental process, for instance in the benchmarks in Figure 2. In particular, many of the values in the tested range $[10, 1, 0.1, 0.01]$ led to almost random optimisation behaviour due to the sensitivity of the loss. We picked the best performing value of 0.1 and made this clearer in the appendix.
>
>
> **Recent related work**
>
> The page limit did prevent us from providing a more extensive overview of possibly related work. We have added a more complete related work section to the appendix (Appendix B.1).
>
> **Isolated issues**
>
> Thanks for spotting them! We replaced them as follows:
> - Line 41: “reparametrisation for categorical distributions, which we discuss further in the related work (Section 5)”
>
> - Line 176-177: “Firstly, two synthetic experiments (Section 6.1 and Section 6.2) will be discussed.”
>
> **DVAE experiments**
>
> * The DVAE is a classical benchmark for gradient estimation methods, as it can easily scale in dimensionality [1, 2]. Moreover, the neural components introduce a complex loss landscape to optimise over.
>
> * The encoder architecture indeed has 2 true hidden layers and one output layer, we rephrased the statement as follows:
> Line 233-235: “The encoder component of the network has two dense hidden layers of sizes 384 and 256 ending in a latent 200-dimensional Bernoulli variable.”
> * Two samples were chosen for IndeCateR in order to perform a sample equivalent comparison to RLOO, which requires at least two samples. This information was added to Appendix A.1 (Hyperparameters). It is an interesting question to analyse the sensitivity of IndeCateR to the number of samples in future work.
> * Our likelihood training is treating the pixels as floats (probabilities) between 0.0 and 1.0.
> We have adjusted our explanation by adding the following to Appendix A.2, Modelling:
> “Finally, following the literature, the output of the decoder is interpreted as the logits for 784 binary random variables and optimised using an ELBO loss function, which is an expected value. The correct probabilities are given by the normalised and binarised pixel values of the original image.”
>
>
> * We only applied the same sampling scheme and also did not use any regularisation, as this could further obscure the impact of using different estimates. These details were also added to the appendix (Appendix A.2, Hyperparameters).
>
> [1] "Categorical Reparameterization with Gumbel-Softmax." ICLR (2016).
>
> [2] "VarGrad: a low-variance gradient estimator for variational inference." NeurIPS (2020).
>
>
> # Questions
>
> **Q1: bias/variance Fig. 1**
>
> We followed the methodology proposed in [3], namely the cosine distance between the estimated and true gradient vector was computed to obtain a scalar metric. This follows the intuition that, during optimisation, we are mainly interested in the direction of the computed gradients.
>
>
> [3] "SIMPLE: A Gradient Estimator for k-Subset Sampling." ICLR 2022.
>
> **Q2: comparison between IndeCateR and GS in terms of variance**
>
> That is indeed a good observation, we speculate that the reason lies in the landscape of the loss function. In Figure 1 and 2, the loss is a direct function of the discrete random variables and behaves more erratically for different instantiations of the variables. In Figure 3, the neural decoder of the DVAE ‘smoothens’ the optimisation, which seems to favour GS.
>
> **Q3: Why sampling? Why binary classifier?**
>
> Regarding the question about sampling. In principle, it is possible to formulate this specific addition problem as suggested. However, our setup is more conceptual in nature. The symbolic component in a neural-symbolic system reasons over the domain of each random variable and the probabilities of those domain elements, not over a statistic. We want to showcase that it is possible to apply sampling in combination with gradient estimation to scale probabilistic neural-symbolic inference and learning tasks. These tasks are known to be hard due to their combinatorial nature [4, 5].
>
> Regarding the question about binary classifier. It is possible to formulate the problem as a “regression” task (where the sum is regarded as a real value). However, our purpose is again different here. The experimental setup for 6.4 mimics exactly how inference and learning would be in a neural-symbolic setting [6]. Such systems are trained on example logical statements that are true or false with a given probability, which translates to our binary supervision signal.
>
> [4] "Scallop: From probabilistic deductive databases to scalable differentiable reasoning." NeurIPS (2021).
>
> [5] "A-nesi: A scalable approximate method for probabilistic neurosymbolic inference." arXiv (2023).
>
> [6] "Deepproblog: Neural probabilistic logic programming." NeurIPS (2018).

---

> > ### Comment · Reviewer_Z21m · 2023-08-16
> > **Thanks for your comments! Revisions are appreciated.**
> >
> > My response here is purely to reply to author comments about my original review. (I haven't looked carefully yet at relationships to LEG and other concerns raised by other reviewers, I look forward to engaging on that in the discussion period).
> >
> > I appreciate the detailed engagement with my questions/comments. I look forward to an improved manuscript. Based on this response, **I continue to think the paper is worth accepting.**
> >
> > RE "AI accelerators" meaning really GPUs: Appreciate the fixes. Thanks for the neat pointer to prefix sums.
> >
> > RE hyperparameters of GS : Thanks for the report of practical difficulty in selecting the value. I think steering reader in main paper to further details about this in appendix would be useful
> >
> > RE DVAE experiments: Thanks for the revisions. Your response helps me understand the motivation and reproducibility of these experiments much better.
> >
> > RE Q1: Thanks. Please clarify how you use cosine distance to get a scalar in the revisions
> >
> > RE Q2: Interesting, I would not have expected a neural decoder to somehow favor one method over alternatives. I wonder if this hypothesis could be scrutinized by substituting a linear decoder and seeing if the same behavior occurs
> >
> > RE Q3: OK, please revise accordingly so that future readers understand that you are pursuing a neurosymbolic kind of task where only boolean logical statements are provided to supervise, and that you understand the alternatives I mention are possible but not of interest to your goals. Otherwise I think readers like me will be distracted by the "why not do it this other way?" ideas like I had.

---

### Author Rebuttal · Authors · 2023-08-08

We thank all reviewers for their valuable feedback and want to specifically draw attention to the comparison between CatLog and Local Expectation Gradients (LEG) [1] as brought up by reviewer JDGY. Please find our theoretical analysis that formally proves the differences in the added pdf.

Indeed, LEG and IndeCateR are related. However, CatLog and LEG are two substantially different tricks/methods for the following reasons:

1. LEG does not make full use of the autoregressive parametrisation of the distribution, as it obtains samples (aka pivots) by first instantiating all variables and subsequently performing evaluation through the computation of a weighted average based on the Markov blanket. CatLog instead makes full use of the autoregressive parametrisation by interleaving sampling and evaluation. LEG is based on three distinct stages: sampling, evaluation and weighted averaging. In contrast, CatLog has only two intertwined stages based on sampling and evaluation. Another way/analogy to look at them is that LEG is inspired and related to Gibbs sampling, whereas CatLog is based on ancestral sampling. Please, refer to the detailed analysis of LEG in the attached PDF, which will be included in the Appendix of our paper and also the table underneath summarizing the main differences.

2. The two methods have different computational complexity. Indeed, computing the weighted average in LEG requires an additional $O(D)$ cost, which contributes to an overall computational complexity of $O(ND^2K)$. Therefore, the CatLog-Derivative trick makes a better use of the structure, resulting in improved efficiency over LEG, as scaling only linearly with the number of variables.

3. As reviewer JDGY also mentioned, the sampling per dimension introduced by CatLog is different from the pivot samples used by LEG. This theoretical difference also yields significant practical differences in performance, even in the case of an independently factorising distribution. This difference can be seen in Section 6.4. There, the ‘IndeCateR’ estimator corresponds to pivot samples (LEG), while ‘IndeCateR-D’ draws new samples per dimension (CatLog). IndeCateR-D is the only estimator that can consistently tackle the problem when increasing the dimensionality, showing that CatLog is both theoretically and practically different from LEG.

| Name  |  Trick | Computational Complexity | Relation to Sampling |
|---|---|---|---|
| LEG  | $\sum_{d=1}^D E_{(X_{<d},{\color{red}X_d'},X_{>d})\sim p(X_{<d},{\color{red}X_d'},X_{>d})} \{ E_{X_d\sim {\color{blue}p(X_d\mid X_{\neq d})}}[f(X)\partial_\lambda \log p(X_d\mid X_{<d})] \}$  | $O(ND^2K)$  | Gibbs sampling  |
| CatLog  | $\sum_{d=1}^D E_{(X_{<d},{\color{red}X_d},X_{>d})\sim p(X_{<d},{\color{red}X_d},X_{>d})} \{ [f(X_{\neq d}, X_d)\partial_\lambda \log p(X_d\mid X_{<d})] \}$ |  $O(NDK)$ | Ancestral sampling |

[1] Titsias, M.K., & Lázaro-Gredilla, M. (2015). Local expectation gradients for black box variational inference. NeurIPS.

---

### Decision · Program_Chairs · 2023-09-21

**Decision:**

Accept (poster)

**Comment:**

Reviewers agreed that this paper is a nice contribution. The relationship to LEG needs to be clarified early in the paper.